



**Uncertainties in temperature statistics and fluxes determined by**
**sonic anemometer due to wind-induced vibrations of mounting arms**
Zhongming Gao[1,2,3], Heping Liu[3,*], Dan Li[4], Bai Yang[5], Von Walden[3], Lei Li[1,2], Ivan Bogoev[5]
[1] School of Atmospheric Sciences, Sun Yat-sen University, Southern Marine Science and
Engineering Guangdong Laboratory (Zhuhai), Zhuhai, China
[2] Key Laboratory of Tropical Atmosphere-Ocean System, Ministry of Education, Sun Yat-sen
University, Zhuhai, Guangdong, China
[3] Laboratory for Atmospheric Research, Department of Civil and Environmental Engineering,
Washington State University, Pullman, Washington, USA
[4] Department of Earth and Environment, Boston University, Boston, Massachusetts, USA
[5] Campbell Scientific, Inc., Logan, Utah, USA
*Corresponding to:  Heping Liu (heping.liu@wsu.edu)
**Abstract**
Accurate air temperature measurements are essential in eddy covariance systems, not only
for determining sensible heat flux but also for applying the density effect corrections (DEC) to
water vapor and $CO_2$ fluxes. However, the influence of wind-induced vibrations of mounting
structures on temperature fluctuations remains a subject of investigation. This study examines 30-
min average temperature variances and fluxes using eddy covariance systems, combining
Campbell Scientific Anemometer Thermometry (CSAT3B) with closely co-located fine-wire
thermocouples alongside LI-COR $CO_2$/$H_2O$ gas analyzers at multiple heights above a sagebrush
ecosystem. The variances of sonic temperature after humidity corrections ($T_s$) and sensible heat
fluxes derived from $T_s$ are underestimated (e.g., by approximately 5% for temperature variances
and 4% for sensible heat fluxes at 40.2 m, respectively) as compared with those measured by a
fine-wire thermocouple ($T_c$). Spectral analysis illustrates that these underestimated variances and





fluxes are caused by the lower energy levels in the $T_s$ spectra than the $T_c$ spectra in the low
frequency range (natural frequency < 0.02 Hz). This underestimated $T_s$ spectra in the low
frequency range become more pronounced with increasing as wind speeds, especially when wind
speed exceeds 10 m s$^{-1}$. Moreover, the underestimated temperature variances and fluxes cause
overestimated water vapor and $CO_2$ fluxes through DEC. Our analysis suggests that these
underestimations when using $T_s$ are likely due to wind-induced vibrations affecting the tower and
mounting arms, altering the time of flight of ultrasonic signals along three sonic measurement
paths. This study underscores the importance of further investigations to develop corrections for
these errors.

**Keywords: Eddy covariance, $CO_2$ fluxes, Fine-wire thermocouple, Sonic temperature; High**
**winds**


**1. Introduction**

The eddy covariance (EC) technique has been widely used to measure turbulent fluxes of

heat, water vapor, $CO_2$, and other scalars between terrestrial ecosystems and the atmosphere (Chu
et al., 2021; Lee et al., 2014; Missik et al., 2021; Tang et al., 2019; Wang et al., 2010). It is
instrumental in studying micrometeorological processes in the atmospheric surface layer (Eder et
al., 2013; Gao et al., 2018; Guo et al., 2009; Li et al., 2018; Zhang et al., 2010). Despite
considerable advancement in the EC technique (Burns et al., 2012; Frank et al., 2013; Fratini et
al., 2012; Horst et al., 2015; Liu et al., 2001; Mauder et al., 2007; Mauder and Zeeman, 2018;
Wilczak et al., 2001), uncertainties in EC fluxes remain a great concern (Loescher et al., 2005;
Massman and Clement, 2006; Peña et al., 2019), including the notable issue of the surface energy
balance closure (Mauder et al., 2020).  Thus, improving the accuracy of EC flux measurements
and identifying the potential sources of uncertainties in these fluxes are critically important.

In most EC applications, sonic-derived air temperature after corrections is usually

employed for determining sensible heat fluxes (H) (Liu et al., 2001; Schotanus et al., 1983).
However, erroneous H determined by sonic anemometers have been reported especially under high
wind conditions (e.g., Burns et al., 2012; Smedman et al., 2007). For instance, Smedman et al.
(2007) utilizing two co-located Gill sonic anemometers (Models R2 and R3) observed that sonic-





determined H exhibited larger magnitudes than H measured with an alternative temperature sensor.
They also noted that for wind speed exceeding 10 m s$^{-1}$, a correction highly dependent on wind
speed is essential for sonic-determined H (Smedman et al., 2007). Burns et al. (2012), employing
a Campbell Scientific sonic anemometer (Model CSAT3) and a co-located type-E thermocouple
(wire diameter of 0.254 mm), reported substantial errors for H determined from the CSAT3 sonic
anemometer with a firmware of version 4.0 for wind speed above 8 m s$^{-1}$. Such large errors in H
result from inaccurate sonic-derived temperature due to an underestimation of the speed of sound,
though errors caused by sonic anemometer transducer shadowing can also cause errors in H (Frank
et al., 2013; Horst et al., 2015). Wind-induced vibrations in the tower and mounting arms,
particularly under windy conditions, were speculated to be potential contributors, causing spikes
in the signals of sonic temperature (Burns et al., 2012). However, the precise impact of vibration-
induced errors in sonic-derived temperature on temperature variances and sensible heat fluxes,
especially for tall towers under strong wind conditions, has remained unexplored.
Accurate air temperature measurements are not only important for determining H but also
crucial for estimating other scalar fluxes (e.g., water vapor and $CO_2$) through density effect
corrections (DEC hereafter; Detto and Katul, 2007; Gao et al., 2020; Lee and Massman, 2011;
Sahlée et al., 2008; Webb et al., 1980). The measured high-frequency time series of densities of
water vapor, $CO_2$, and other scalars are subjected to the effects of density fluctuations of dry air
and other components in the atmosphere, as well as the fluctuations of air pressure (Lee and
Massman, 2011; Webb et al., 1980). Correcting for these effects involves applying corrections to
either the calculated raw fluxes or to the high-frequency time series of the scalar density
fluctuations (Webb et al., 1980; Detto and Katul, 2007; Gao et al., 2020; Sahlée et al., 2008). Any
errors or uncertainties in sonic-derived temperature are anticipated to propagate, and in some
certain cases, be amplified by the correction algorithms applied to the scalar fluxes, leading to
heightened uncertainties in these fluxes (Liu et al., 2006).
The objective of this study Is to scrutinize the uncertainties in temperature statistics and
fluxes determined by sonic anemometers, with particular attention to the potential influence of
vibrations of the tower and mounting arms under high wind speeds. The data employed were
collected from three levels of Campbell Scientific sonic anemometers (Model CSAT3B) alongside
co-located fine wire thermocouples and open-path infrared gas analyzers. By comparing the
sensible heat fluxes calculated using air temperature from the sonic anemometers and the





thermocouples, we assess vibration-induced errors in sensible heat fluxes at the three heights. The
findings reveal that the sonic anemometers underestimate the temperature variances and fluxes
compared to the thermocouples. Furthermore, we investigate the propagation of these vibration-
induced errors to water vapor and $CO_2$ fluxes through the density effect corrections.

**2. Materials and methods**
**2.1 Experiment and data**
The experiment was conducted at the National Oceanic and Atmospheric Administration
(NOAA) Grid 3 area (Station ID: GRI) situated on the western edge of the Snake River Plain in
southeastern Idaho, USA (43.59°N, 112.94°W; 1,500 m above mean sea level; Figure 1). The
closest mountains are located approximately 13 km northwest from GRI. Based on the data from
multiple automated meteorological observation stations in the area, southwesterly and
northeasterly winds prevail during the day and night, respectively (Finn et al., 2016). Under these
prevailing winds, GRI has a relatively flat and uniform upwind fetch (Finn et al., 2018; Lan et al.,
2018). The vegetation primarily comprises shrubs and grasses, each with a roughness length and
displacement height of a few centimeters (Finn et al., 2016).
The experiment utilized a 62-m tower and a 10-m tower at the Grid 3 area to mount the
sensors (Figure 1). The 62-m tower was guyed at eight levels and the 10-m tower was guyed at
one level. 3.6m (12-ft) retractable booms were horizontally braced to the 62-m tower to attach the
CSAT3s and IRGAs. These sensors were mounted at the end of the booms, and the CSAT3s were
well-aligned to the booms. As a result, the CSAT3s and IRGAs were positioned at least 2.0 m
away from the 62-m tower. On the 10-m tower, 1.8 m (6-ft) poles were utilized to mount the
CSAT3s and IRGAs, positioning the sensors approximately 1.5 m away from the tower's frame
structures.




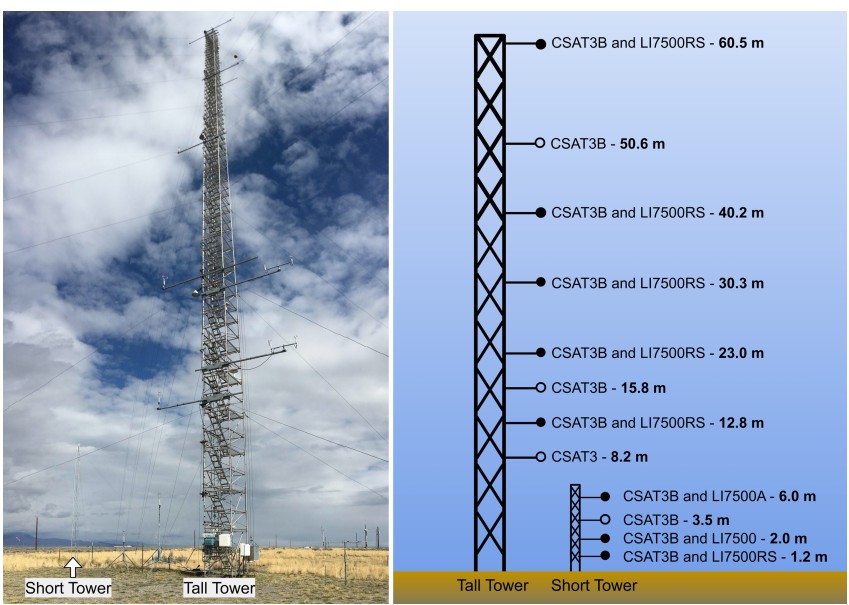

**Figure 1.** Photos of the 62-m and 10-m towers at the Idaho National Laboratory (INL) site in
southeastern Idaho, and the instrumentational configuration of the field study.

Throughout the experiment, multiple levels of EC systems were deployed. These included
two models of 3D sonic anemometers from the same manufacturers (Model CSAT3B and CSAT3,
Campbell Scientific, Inc.) and three models of infrared gas analyzers from the same manufacturers
(IRGA; Model LI7500RS, LI7500A, and LI7500, LICOR, Inc.). CSAT3s measured the three-
dimensional wind velocity components ($u$, $v$, and $w$) and the sonic air temperature ($T_{s\_m}$), while
IRGAs measured the densities of water vapor ($\rho_v$) and $CO_2$ ($\rho_c$). Co-located with CSAT3s and
IRGAs, Type-E fine wire thermocouples (Model FW3, Campbell Scientific, Inc.) were used to
measure air temperature ($T_c$). Specifically, the FW3 thermocouple is composed of a chromel wire
and a constantan wire with diameters of 0.0762 mm. The FW3 determines $T_c$ by measuring the
voltage potential differences created at the junction of the two wires due to the temperature
difference, whereas $T_{s\_m}$ is determined based on the relationship between sonic virtual temperature
and the speed of sound (Liu et al., 2001). This distinction in temperature measurement principles
between CSAT3s and FW3 implies that the measured $T_{s\_m}$ and $T_c$ are independent of each other.
Furthermore, $T_c$ is expected to remain unaffected by vibrations of the tower and mounting arms.
In this study, we also recorded the inclination of the CSAT3B (e.g., pitch and roll angle





measurements) given by an integrated inclinometer in the CSAT3B. Pitch angle is defined as the
angle between the gravitationally horizontal plane and the CSAT3B x-axis, and roll angle is
defined as the angle between the gravitationally horizontal plane and the CSAT3B y-axis. The
sonic roll and pitch angles were stored as 30-min averages.

Three dataloggers (Model CR1000X, Campbell Scientific, Inc.) were employed to sample

the high-frequency instruments (i.e., sonic anemometer, fine-wire thermocouple, and gas analyzer)
at 10 Hz. Each datalogger is equipped with a GPS receiver (Model GPS16X-HVS, Garmin
International, Inc.) to synchronize the datalogger clocks. Additionally, a variety of meteorological
measurements were conducted at GRI, including net radiation, air temperature, relative humidity,
soil moisture, soil temperature, and soil heat fluxes. The data examined in this study spans from
25 April to 31 July 2021, collected at three heights: 40.3, 23.0 and 12.8 m.

**2.2 Post-field data processing**

The data processing mainly entailed despiking, double rotation for the wind components

(Wilczak et al., 2001), sonic temperature conversion (Liu et al., 2001; Schotanus et al., 1983), and
application of DEC to the raw fluxes of latent heat and $CO_2$ (Webb et al., 1980). However,
corrections for the effects of humidity and density fluctuations in this study are applied to the
turbulent fluctuations of sonic temperature, $\rho_v$, and $\rho_c$, respectively (Detto and Katul, 2007; Gao
et al., 2020; Sahlée et al., 2008; Schotanus et al., 1983; Webb et al., 1980). For each 30-min interval,
the corrected turbulent fluctuations of sonic temperature ($T_s'$), $\rho_v'$, and $\rho_c'$ are determined by,

$$T_s'(t) = T_{s\_m}'(t) - 0.51\frac{\rho_v'(t)}{\bar{\rho}_a}\bar{T}, \tag{1}$$

$$\rho_v'(t) = (1 + \mu\sigma)\rho_{v\_m}'(t) + (1 + \mu\sigma)\frac{\bar{\rho}_v}{\bar{T}}T_s'(t), \tag{2}$$

$$\rho_c'(t) = \rho_{c\_m}'(t) + \bar{\rho}_c(1 + \mu\sigma)\frac{T_s'(t)}{\bar{T}} + \mu\frac{\bar{\rho}_c}{\bar{\rho}_a}\rho_v'(t). \tag{3}$$

where $T_{s\_m}$, $\rho_{v\_m}$, and $\rho_{c\_m}$ represent measured sonic temperature, water vapor and $CO_2$ densities,
respectively; $\bar{T}$, $\bar{\rho}_a$, $\bar{\rho}_v$, and $\bar{\rho}_c$ are averages of air temperature, air density, water vapor and $CO_2$
densities, respectively; $\mu = m_d/m_v$ ($m_d$ and $m_v$ are the molecular mass of dry air and water
vapor, respectively); $\sigma = \bar{\rho}_v/\bar{\rho}_d$ ($\bar{\rho}_d$ is the density of dry air). The prime symbol denotes the





turbulent fluctuations relative to the 30-min block average. As shown in the equations above, there
is interdependence between $T_s'$ and $\rho_v'$, and thus $T_s'$ and $\rho_v'$ must be determined iteratively. In this
study, the corrections are iterated twice. Note that in semiarid sites like ours, the adjustments to $T_s'$
due to fluctuations in specific humidity ($\frac{\rho_v'(t)}{\bar{\rho}_a}$) are typically negligible (not shown here) as
compared to the adjustment to $\rho_v'$ due to fluctuations in $T_s'$ (as demonstrated in Section 3.5). The
corrected time series of fluctuations facilitate the investigation of coherent structures and scalar
similarity between temperature and other scalars (Detto and Katul, 2007; Sahlée et al., 2008). Here
and throughout, $T_s$ refers to the air temperature measured by sonic anemometers after humidity
corrections, $T_{s\_m}$ the sonic temperature directly measured before corrections, and $T_c$ the air
temperature measured by fine-wire thermocouples (FW3).

**2.3 Ensemble empirical mode decomposition (EEMD)**
The ensemble empirical mode decomposition (EEMD) (Huang et al., 1998; Huang and Wu,
2008) is applied to decompose the 30-min turbulence time series into three subsequences,
corresponding to the high, middle, and low frequency ranges, respectively. EEMD is a favored
method in analyzing non-linear and non-stationary turbulence data (Gao et al., 2018; Hong et al.,
2010; Liu et al., 2021; Huang et al., 1998; Huang and Wu, 2008). Through the sifting process in
EEMD, a 30-min time series is decomposed into thirteen oscillatory components $C_j(t)$ (j = 1, 2, …,
13) and an overall residual $r_{13}(t)$. Each oscillatory component generally exhibits one
characteristic frequency (Hong et al., 2010; Gao et al., 2018), while the overall residual is either
monotonic or containing only one extremum, from which no more oscillatory components can be
further decomposed. Hence,

$$x(t) = r_{13}(t) + \sum_{j=1}^{13} C_j(t). \tag{4}$$

As detailed in Section 3.2, after comparing the power spectra of $T_s$ and $T_c$, two frequency
boundaries, 0.02 and 0.2 Hz, are identified. The oscillatory components are then categorized into
three regimes (I, II, and III, respectively). The oscillatory components with mean frequencies
falling within the corresponding ranges are added together to generate the three subsequences.
Specifically, oscillatory components with the mean frequencies smaller than 0.02 Hz are summed
and labeled as regime I (i.e., $x_I' = \sum_{j=10}^{13} C_{j,x} + r_{13}$, where $x = w, T_s$, and $T_c$), between 0.02 Hz and





0.2 Hz as regime II (i.e., $x'_{II} = \sum_{j=6}^{9} C_{j,x}$), and larger than 0.2 Hz as regime III (i.e., $x'_{III} =$
$\sum_{j=1}^{5} C_{j,x}$).

**3 Results and discussion**
**3.1 Comparison of the CSAT3B- and FW3-derived temperature variances and fluxes**
Figure 2 illustrates the comparisons between the variances and fluxes obtained using $T_s$
and $T_c$ at different heights. Here, we use $\sigma_{T_c}^2$ and $\overline{w'T_c'}$ as reference values since $T_c$ is less sensitive
to the effects of humidity and wind speeds than $T_s$ (Burns et al., 2012; Smedman et al., 2007).
Generally, the variances of $T_s$ ($\sigma_{T_s}^2$) are smaller than variances of $T_c$ ($\sigma_{T_c}^2$), typically by 2%–5%
(Figures 2a-2c). These lower variances result in a lower sensible heat flux ($\overline{w'T_s'}$). Specifically, at
23.0 and 40.2 m, $\overline{w'T_s'}$ is underestimated by approximately 2% and 4%, respectively, as compared
to the sensible heat fluxes derived from the FW3 (i.e., $\overline{w'T_c'}$) (Figures 2d and 2e). However, at 12.8
m, $\overline{w'T_s'}$ and $\overline{w'T_c'}$ are quite comparable (Figure 2f). These results confirm that errors in H were
not entirely due to errors in the vertical velocity (Frank et al., 2013; Horst et al., 2015). Further
examination of Figure 2 reveals that the differences between $\sigma_{T_s}^2$ and $\sigma_{T_c}^2$, as well as between $\overline{w'T_s'}$
and $\overline{w'T_c'}$, increase with the increasing measurement heights. Additionally, our tests indicate that
the influence of solar heating on measurements of FW3 is negligible, primarily due to the thin wire
diameter of 0.0762 mm (Text S1 and Figure S1).

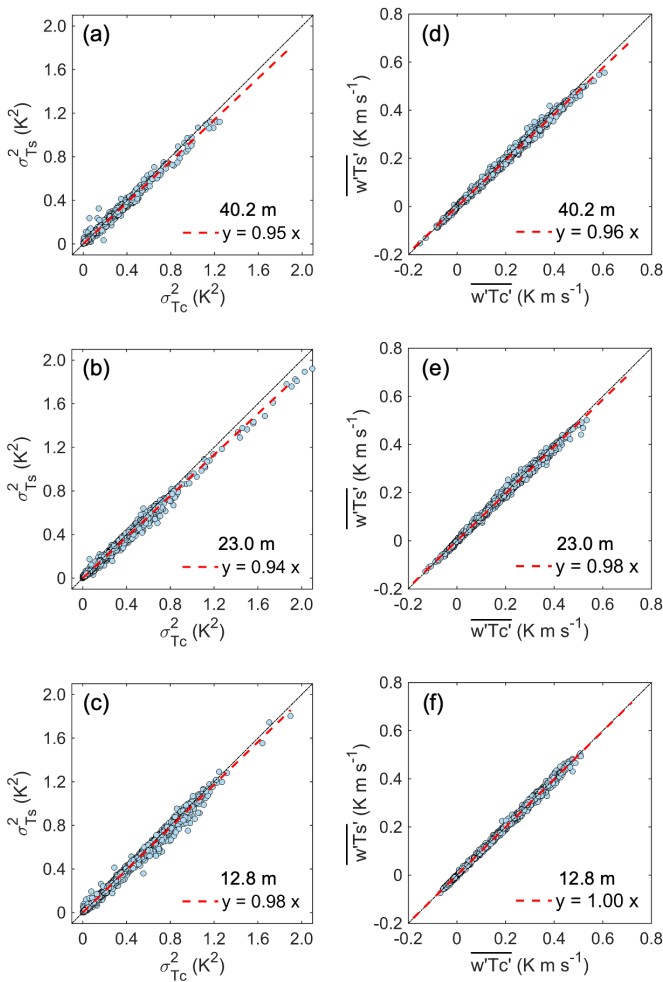

**Figure 2.** Comparison of temperature variances and sensible heat fluxes computed using $T_s$ and $T_c$ at the heights of 40.2, 23.0 and 12.8 m, respectively.

What mechanisms could have caused the observed differences? Previous studies found that CSAT3 sonic anemometers with a previous version of firmware could cause errors in $T_s$ (Burns et al., 2012). However, this should not be the case for our study because the CSAT3B modes were used at these heights and they have an improved design compared to the original CSAT3. Our results suggest that the differences are dependent on the measurement heights. As the measurement heights increase, the dominant length scale of coherent structures is also enlarged (Zhang et al.,





2011). Therefore, we conjecture that the processes that lead to such differences are most likely
scale dependent, motivating us to examine the spectra and cospectra in the next subsection.

**3.2 Spectral comparisons**
To gain further insight into the differences between $\sigma_{T_s}^2$ and $\sigma_{T_c}^2$ as well as the associated
fluxes (i.e., $\overline{w'T_s'}$ and $\overline{w'T_c'}$), the spectra of $u$, $w$, $T_s$ and $T_c$ and the $w$-$T_s$ and $w$-$T_c$ cospectra at
different heights are examined. Figures 3a-3c show the mean normalized Fourier power spectra of
$u$, $w$, $T_s$, and $T_c$ as a function of natural frequency ($f$). Note that the power spectra were computed
every half hour using fast Fourier transform and normalized by the corresponding variances before
averaging. It is also interesting to note that the $u$, $w$, and $T_s$ spectra deviate from the well-known
$-5/3$ power law in the high frequency range of $f > 0.2$ Hz, exhibiting similar features as previous
studies (e.g., Burns et al., 2012). For $f > 2$ Hz, $fS_u$, $fS_w$, and $fS_{Ts}$ appear to follow the $f^{+1}$ slope,
likely due to white noise and/or aliasing (Kaimal and Finnigan, 1994). In the 0.2 Hz < $f$ < 2 Hz
range, the distortion of $fS_u$, $fS_w$, and $fS_{Ts}$ from the $-5/3$ power law is enhanced as the
measurement heights decrease. The upturned distortion for 0.2 Hz < $f$ < 2 Hz might be associated
with spikes (Gao et al., 2020; Stull, 1988) that are not excluded from the 10 Hz time series during
despiking. The $T_c$ spectra appear to follow the $-5/3$ power law for 0.2 Hz < $f$ < 1 Hz, although are
slightly attenuated for $f > 1$ Hz, likely because the thermal mass of the thermocouple wire limits
its response time (Burns et al., 2012).

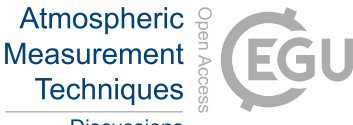

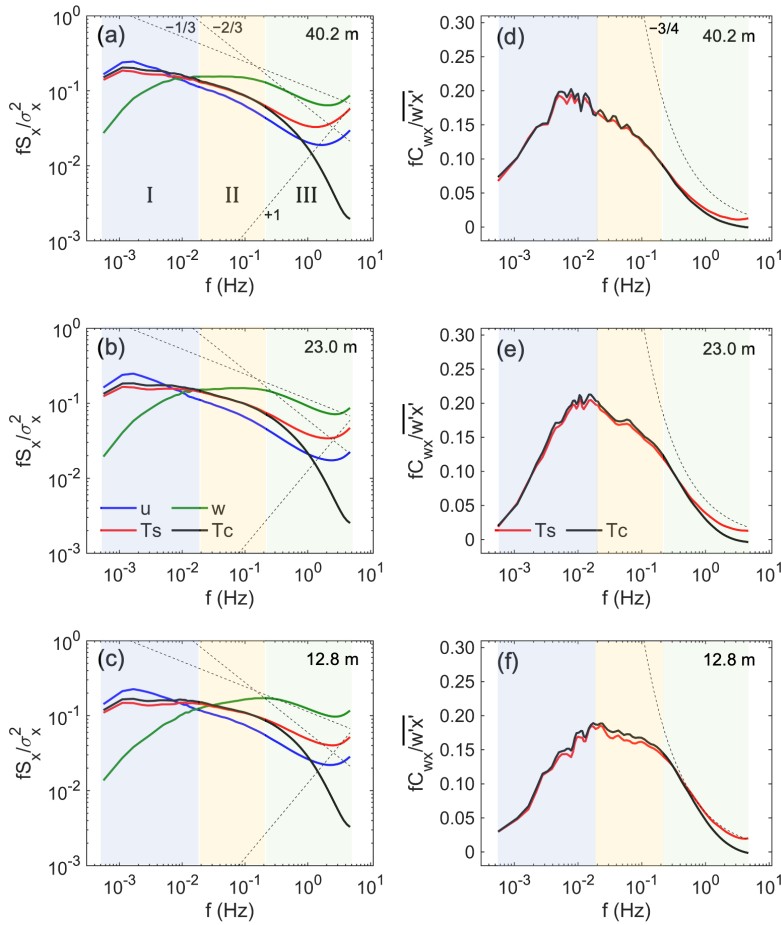

**Figure 3.** Mean normalized power spectra of u, w, $T_s$, and $T_c$, and cospectra of the $w$-$T_s$, and $w$-$T_c$, at three heights of 40.2, 23.0, and 12.8 m, respectively. All power spectra and cospectra are normalized by the corresponding variance and covariance before averaging. The dashed lines show a $f^{-1/3}$, $f^{-2/3}$, $f^{+1}$, and $f^{-3/4}$ slope. Note that the frequency domain can be divided into three regions by comparing the $T_s$ and $T_c$ spectra.

It is also noted that the magnitude of the $T_c$ spectra is higher than that of the $T_s$ spectra in the low frequency range of $f < 0.02$ Hz, but their magnitude is comparable in the middle frequency range of 0.02 Hz $< f <$ 0.2 Hz. These results indicate that turbulent eddies with scales less than 0.02 Hz contribute more to $\sigma_{T_c}^2$ than $\sigma_{T_s}^2$. Hence, the underestimation of $\sigma_{T_s}^2$ is mostly caused by the





lower magnitude of the $T_s$ spectra in the low frequency range, whereas $\sigma^2_{T_s}$ is overestimated to
some extent due to the upturned distortion of the $T_s$ spectra in the high frequency range.

Figures 3d-3f show the distribution of the mean normalized $w$-$T_s$ and $w$-$T_c$ cospectra as a

function of $f$. Each cospectra is normalized by the corresponding covariances before averaging.
For $f > 0.2$ Hz, especially when $f > 0.5$ Hz, the $w$-$T_s$ cospectra are generally higher than the $w$-
$T_c$ cospectra, consistent with the higher spectra of $T_s$. Compared to the power law of –3/4, the
calculated $\overline{w'T_s'}$ are overestimated, while $\overline{w'T_c'}$ is slightly underestimated. For 0.002 Hz $< f < 0.2$
Hz, the normalized $w$-$T_s$ cospectra are slightly lower than the normalized $w$-$T_c$ cospectra,
indicating that turbulent eddies with scales in the frequency range of 0.002 Hz $< f < 0.2$ Hz
contribute more to $\overline{w'T_c'}$. Overall, the underestimation of the CSAT3B-derived fluxes is also scale-
dependent, and the underestimation of $\overline{w'T_s'}$ in the middle to low frequency range is offset by the
overestimation in the high frequency range to some extent.

According to the comparison of the $T_s$ and $T_c$ spectra, the whole frequency domain can be

divided into three regimes: I) $f < 0.02$ Hz, II) 0.02 Hz $< f < 0.2$ Hz, and III) $f > 0.2$ Hz. For $f <$
0.02 Hz, the magnitude of the $T_c$ power spectra is slightly higher than that of the $T_s$ spectra at all
levels, but the magnitude of the $T_s$ and $T_c$ spectra is comparable in the middle frequency range of
0.02 Hz $< f < 0.2$ Hz. For $f > 0.2$ Hz, the $T_c$ spectra first follow the $-5/3$ power law and are then
attenuated, whereas the $T_s$ spectra are distorted upward. Based on this division, in section 3.3, the
30-min time series is divided into the three regimes and the contributions of these different scales
to $\sigma^2_{T_s}$, $\sigma^2_{T_c}$, $\overline{w'T_s'}$, and $\overline{w'T_c'}$ at different heights are then quantified.

### 3.3 Scale-dependent contributions to variances and fluxes

To quantify the contributions of different scales to the corresponding temperature variances

and fluxes, we apply the EEMD approach to decompose the 30-min time series of $w'$, $T_s'$, and $T_c'$
into various oscillatory components, which are then categorized into the three regimes discussed
earlier (i.e., I, II, and III). Figures 4a-4c depict that for regime I, the ratios between the variances
of $T_s$ and $T_c$ are generally lower than 1.0 (approximately 0.89 on average). This suggests that
turbulent eddies with scales less than 0.02 Hz contribute approximately 11% more to $\sigma^2_{T_{c,I}}$ than
$\sigma^2_{T_{s,I}}$. With these turbulent eddies contributing about 41%–57% to the total variances (Figures 5a-
5c and Table 1), the 11% difference between $\sigma^2_{T_{c,I}}$ and $\sigma^2_{T_{s,I}}$ would cause 4%–6% difference





between the total variances of $T_s$ and $T_c$. As for fluxes, turbulent eddies with scales less than 0.02
Hz contribute approximately 6% more to $\overline{w'T_s'}_I$ than $\overline{w'T_c'}_I$. With these turbulent eddies
accounting for about 26%–45% of the total fluxes (Figures 5d-5f and Table 1), the 6% difference
in $\overline{w'T_s'}_I$ and $\overline{w'T_c'}_I$ would cause 2%–3% difference in the total fluxes. Further, given that the
contribution of regime I to the total temperature variances and fluxes increases with measurement
height (Figure 5 and Table 1), the underestimation becomes more significant at higher levels.

**Table 1.** Contributions of the three regimes to the total temperature variances and fluxes of $T_s$ and
$T_c$, as well as the mean ratios of temperature variances and fluxes of $T_s$ and $T_c$ for each regime.

|  |  | 40.2 m | 23.0 m | 12.8 m |
|---|---|---|---|---|
| $\sigma^2_{Ts,i}\big/\sigma^2_{Ts}$ | I | 0.51 | 0.46 | 0.41 |
|  | II | 0.32 | 0.36 | 0.37 |
|  | III | 0.17 | 0.18 | 0.22 |
| $\sigma^2_{Tc,i}\big/\sigma^2_{Tc}$ | I | 0.57 | 0.52 | 0.47 |
|  | II | 0.34 | 0.36 | 0.38 |
|  | III | 0.10 | 0.12 | 0.15 |
| $\sigma^2_{Ts,i}\big/\sigma^2_{Tc,i}$ | I | 0.89 | 0.88 | 0.89 |
|  | II | 0.98 | 0.98 | 0.99 |
|  | III | 1.49 | 1.42 | 1.44 |
| $\overline{w'Ts'}_i\big/\overline{w'Ts'}$ | I | 0.43 | 0.34 | 0.26 |
|  | II | 0.42 | 0.47 | 0.48 |
|  | III | 0.14 | 0.19 | 0.26 |
| $\overline{w'Tc'}_i\big/\overline{w'Tc'}$ | I | 0.45 | 0.36 | 0.27 |
|  | II | 0.43 | 0.48 | 0.50 |
|  | III | 0.12 | 0.16 | 0.23 |
| $\overline{w'Ts'}_i\big/\overline{w'Tc'}_i$ | I | 0.94 | 0.94 | 0.95 |
|  | II | 0.98 | 0.99 | 0.99 |
|  | III | 1.17 | 1.23 | 1.19 |





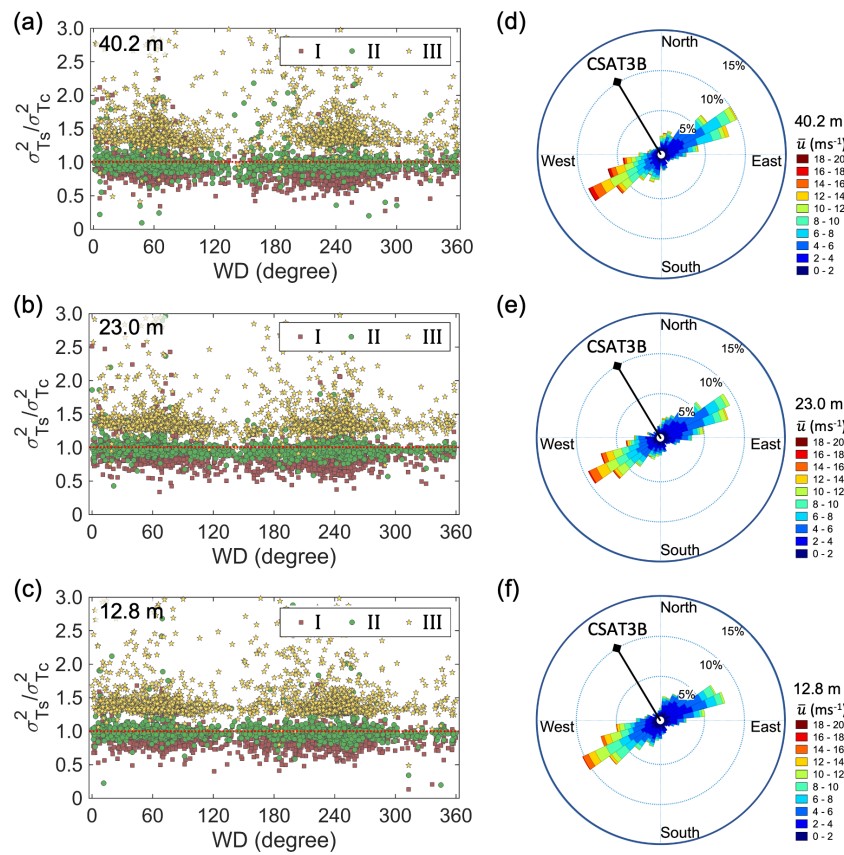


**Figure 4.** (a-c) Distribution of ratios of half-hourly variances of $T_s$ and $T_c$ with wind directions

and (d-f) wind roses at the heights of 40.2, 23.0, and 12.8 m, respectively. The square, circle, and

pentagram markers represent the ratios of the variances in the three regimes. The black lines in d-

f refer to directions of the mounting arms of instruments.




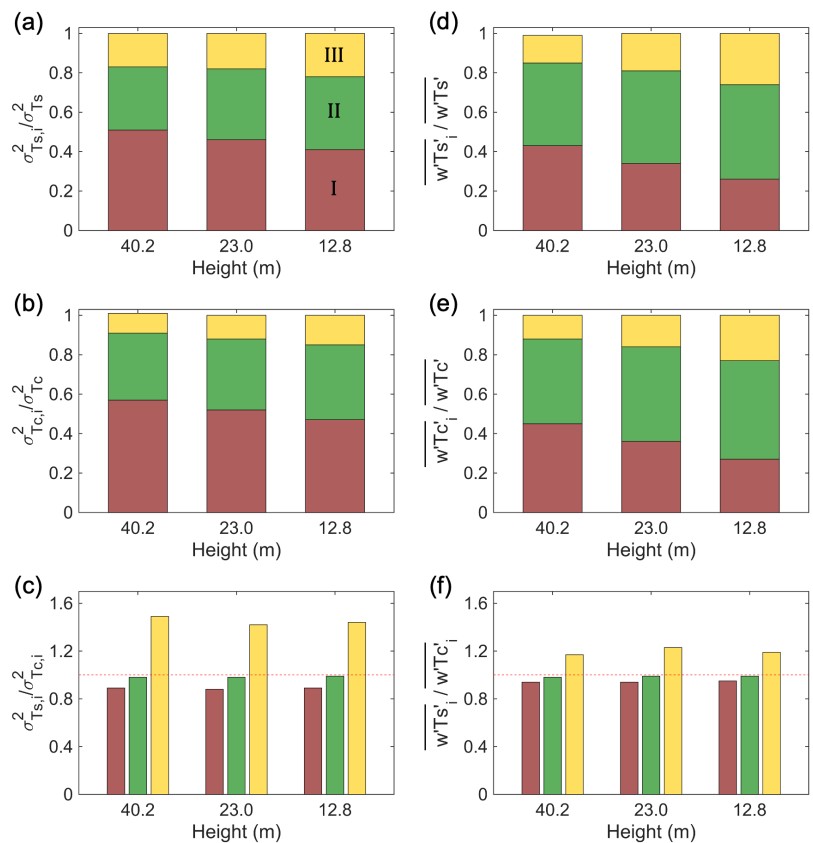


**Figure 5.** Contributions of the three regimes to the total temperature variances and fluxes of $T_s$

and $T_c$, respectively, as well as the mean ratios of the temperature variances and fluxes of $T_s$ and

$T_c$ for each regime.


For regime II, the ratios between the variances of $T_s$ and $T_c$ and the associated fluxes are
close to 1.0 with minimal scatter. This indicates that turbulent eddies with scales between 0.02 Hz
and 0.2 Hz contribute consistently to the temperature variances and fluxes. These turbulent eddies
contribute 32%–38% and 42%–50% to their total variances and fluxes, respectively (Figure 5 and
Table 1). For regime III, the ratios between the variances of $T_s$ and $T_c$ average about 1.45 at the
three heights due to the distorted spectra in this regime for both $T_s$ and $T_c$. This indicates that
turbulent eddies with scales larger than 0.2 Hz contribute roughly 45% less to $\sigma_{T_c,III}^2$ than $\sigma_{T_s,III}^2$.
With these turbulent eddies contributing about 12%–26% to the total variances (Figures 5a-5c and
Table 1), the 45% difference in $\sigma_{T_c,III}^2$ and $\sigma_{T_s,III}^2$ would cause 5%–7% difference in the total





variances of $T_s$ and $T_c$. As for fluxes, turbulent eddies with scales larger than 0.2 Hz contribute
approximately 20% less to $\overline{w'T_c'}_{III}$ than $\overline{w'T_s'}_{III}$. With these turbulent eddies accounting for 12%–
26% of the total fluxes (Figures 5d-5f and Table 1), the 20% difference in $\overline{w'T_c'}_{III}$ and $\overline{w'T_s'}_{III}$
would cause 2%–4% difference in the total fluxes. These results suggest that the observed
underestimation of the $T_s$ variances and fluxes is primarily attributed to the large turbulent eddies
with frequencies less than 0.02 Hz, which is offset to some extent by the contribution from small
turbulent eddies with frequencies larger than 0.2 Hz.

**3.4 Potential causes for the scale-dependent differences**

The potential causes for the scale-dependent differences between the $T_s$ and $T_c$ spectra

include measurement errors, solar heating of thermocouples, and tower and mounting arm
vibrations, among others (e.g., the deficiency in the design of sonic anemometers in response to
different wind speed conditions). The $T_c$ spectra follow similar declining features in the high
frequency range independent to wind speed (Text S2 and Figure S2), suggesting that
measurements of the fine-wire thermocouples were not noticeably affected by increased wind
speed, and therefore operational errors could be excluded from the causes for the observed
difference. Additionally, the consistent differences between the power spectra of $T_s$ and $T_c$ under
nighttime and daytime conditions (Text S1 and Figure S1) suggest that the impact of solar heating
on thermocouples was also not the cause for the differences between the $T_s$ and $T_c$ variances and
fluxes.

As the wind speed increases, the tower vibrations become more pronounced, especially at

higher levels, as indicated by the more energetic peaks in the power spectra of sonic roll and pitch
angles (Figure 6). For wind speed below approximately 10 m s$^{-1}$ at 40.2 m, the ratios of the $T_s$ and
$T_c$ variances for regime I show no obvious change with wind speed. However, for wind speed
above 10 m s$^{-1}$, the ratios decrease further as wind speed increases (Figure 7a). The ratios of the
temperature fluxes of $T_s$ and $T_c$ exhibit a similar pattern to the variances (Figure 7d). For regime
II, the ratios of both the variance and fluxes of $T_s$ and $T_c$ show no obvious relations with wind
speed (Figures 7b and 7e). For regime III, the ratios of both the variance and fluxes of $T_s$ and $T_c$
illustrate large scatter, especially when wind speed is below 10 m s$^{-1}$, but still show no obvious
trends as wind speed increases (Figures 7c and 7f).



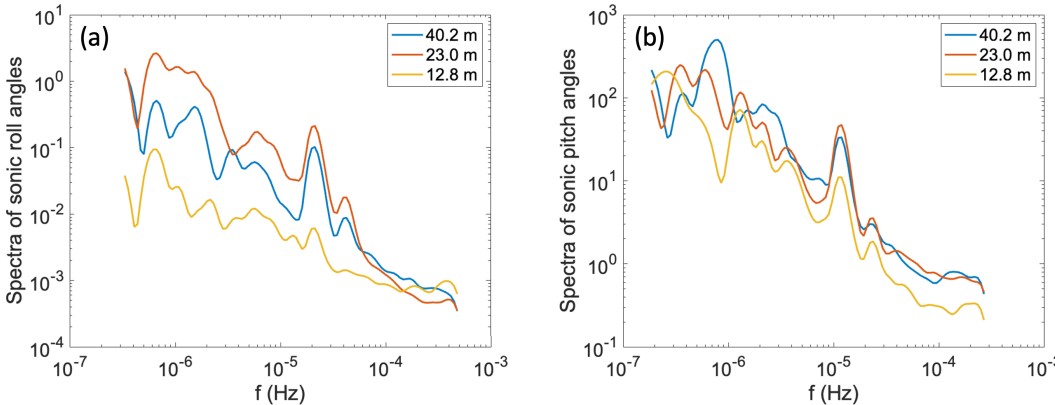

**Figure 6.** Power spectra of the sonic (a) roll and (b) pitch angles measured by CSAT3B at the three heights. The sonic roll and pitch angles were stored as 30-min averages during the experiment.

We thus hypothesize that the enhanced tower vibrations under strong winds lead to an early or delayed detection of the sonic pulse. This results in overestimations or underestimations of the speed of sound and thus errors in the 10 Hz time series of sonic temperature. In this study, for wind speeds below 10 m s$^{-1}$, the ratios of the variances (and fluxes) of $T_s$ to those of $T_c$ are scattered around 1.0 for regime I. However, for wind speed above 10 m s$^{-1}$, the ratios decrease as wind speed further increases (Figure 7). Therefore, tower and mounting arm vibrations were most likely the cause for the differences in the temperature variance and fluxes. Under this circumstance, such vibrations may also affect sonic-measured wind components along with $T_s$, resulting in errors in all the calculated fluxes. However, rigorous tests of this hypothesis seem necessary through testing sonic anemometers in wind tunnels or fields with different mounting strategies.

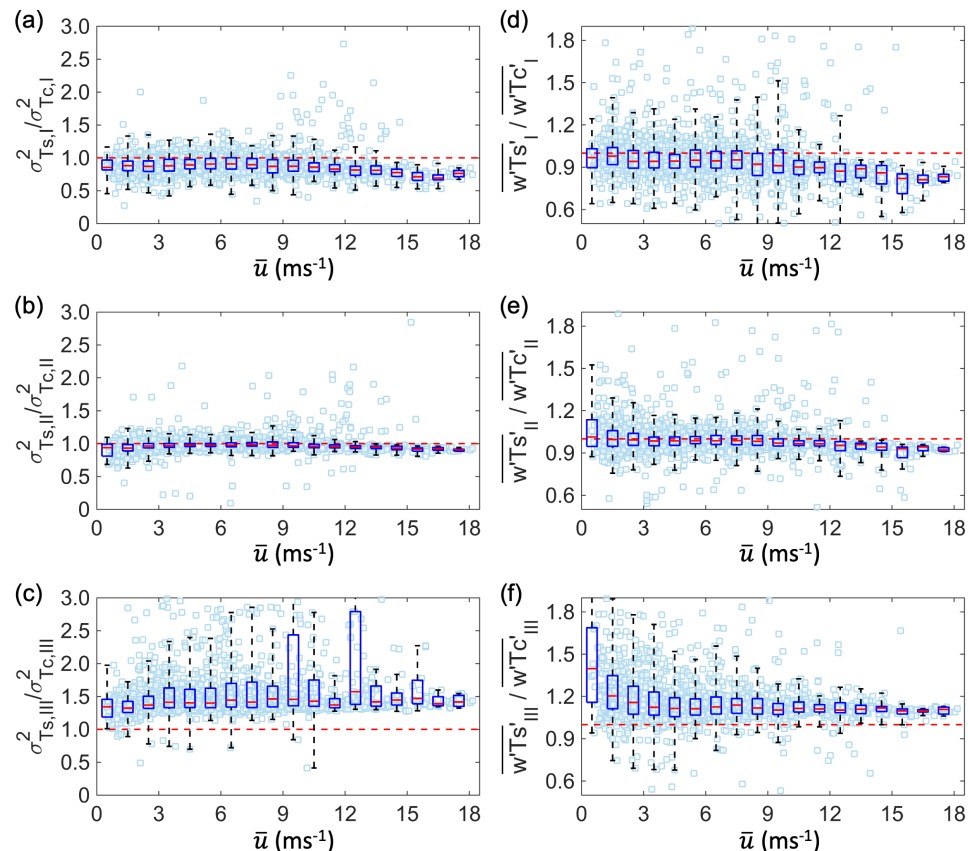

**Figure 7.** Ratios of (a, b, and c) half-hourly variances of $T_s$ and $T_c$ and (d, e, and f) covariances of $w$-$T_s$ and $w$-$T_c$ corresponding to the low, middle, and high frequency ranges, respectively, as a function of the mean wind speed ($\bar{u}$) at 40.2 m.

### 3.5 Implications to water vapor and CO$_2$ fluxes

Given that equations (2) and (3) are used to adjust the measured densities of water vapor and CO$_2$ by open-path CO$_2$/H$_2$O gas analyzers in EC systems, any errors in temperature measurements would be propagated to water vapor and CO$_2$ time series through these two equations. In equations (2) and (3), $T_s$ can be replaced by $T_c$ to achieve the adjusted time series of densities of water vapor and CO$_2$ for FW3. The adjusted time series of water vapor and CO$_2$ by the fluctuating parts of $T_s$ and $T_c$, respectively, are then decomposed into three regimes to quantify the influence of $T_s$ on variances and fluxes of water vapor and CO$_2$.



The variances of $\rho_v$ are not influenced by using either $T_s$ or $T_c$ for the density effect

corrections (Table S1) because the fluctuations in $\rho_v$ are not very sensitive to the density effects,

especially at semiarid sites like ours (Gao et al., 2020). Therefore, for the water vapor fluxes, there

only exist minor differences (< 1%) between the fluxes corrected by $T_s$ and $T_c$ for the three regimes,

while the overall water vapor fluxes corrected by $T_s$ and $T_c$ are comparable at the three heights

(Figures 8a-8c).

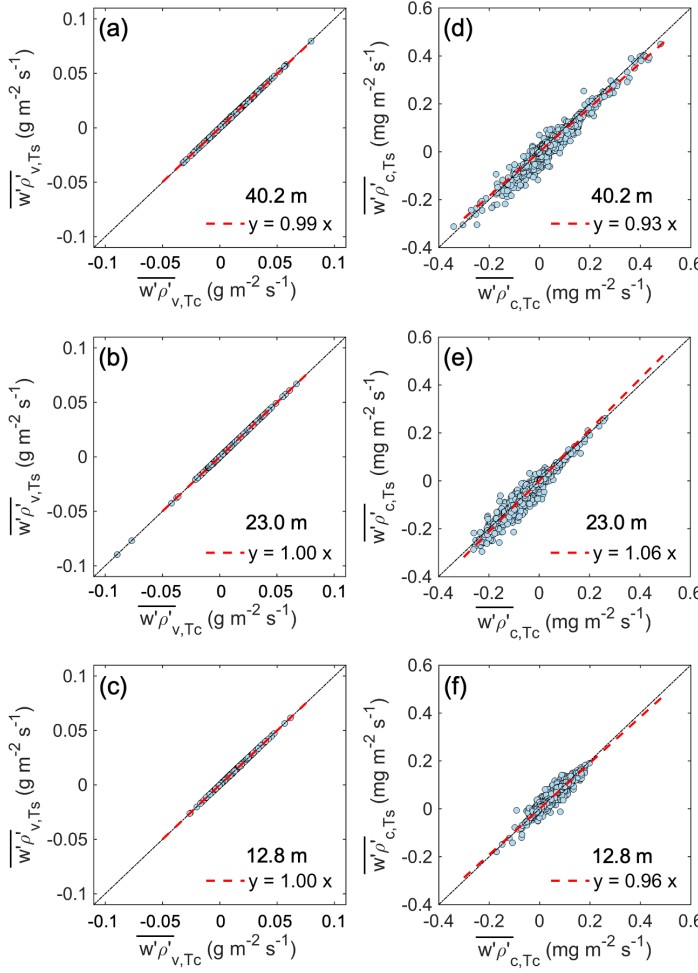


**Figure 8.** Comparison of water vapor and $CO_2$ fluxes corrected by temperature from sonic
anemometers and fine wire thermocouples, respectively, at the three heights.





As for $CO_2$, the differences between $T_s$ and $T_c$ in the three regimes are propagated

differently to the adjusted variances of $\rho_c$ (Table S2). More importantly, there exist relatively large
differences (but still within 10%) between the $CO_2$ fluxes adjusted by $T_s$ and $T_c$ (Figures 8d-8f).
In general, $CO_2$ fluxes are more sensitive to errors or uncertainties in temperature measurements
than water vapor fluxes (Liu et al., 2006). Therefore, precise measurements of air temperature are
critical for quantifying ecosystem $CO_2$ fluxes.


**4 Conclusions**

Temperature variances and the associated fluxes are examined by using sonic anemometers

and co-located fine-wire thermocouples at three levels above a sagebrush ecosystem. Compared
to temperature variances and fluxes determined by thermocouples, sonic anemometers were found
to underestimate the variances and fluxes by approximately 5% and 4%, respectively, at the height
of 40.2 m. In the high frequency range, the distortion in $T_s$ spectra contributed to the heat fluxes
by 2–4%, whereas the attenuation in $T_c$ spectra led to underestimated fluxes. However, the primary
source of the underestimation in temperature variances and fluxes by the sonic anemometers was
identified in the low frequency range. This phenomenon became more profound with increasing
wind speed, and thus, heightened vibrations of the tower and mounting arms. Furthermore, when
adjusting the density effects using the attenuated temperature, our results indicate that $CO_2$
variances and fluxes are more sensitive to errors or uncertainties in temperature measurements
compared to those of water vapor at the semiarid site.

Our findings highlight the critical importance of accurate measurements of air temperature

fluctuations in EC flux measurements. Furthermore, the observed underestimation of sonic
temperature variances and fluxes suggests that the measured wind velocity components may also
be biased due to the tower vibrations. Therefore, we recommend further investigation of the
influence of mounting arm vibrations on wind velocity components by sonic anemometers.

**Data availability**

The data used in this study are available from Heping Liu upon request.




**Code availability**

The code used in this study are available from Zhongming Gao and Heping Liu upon request.

**Author contributions**

ZG and HL designed the study with substantial input from all coauthors. ZG, HL, DL, and BY conducted the fieldwork and obtained and processed the EC data. ZG drafted the manuscript. All authors contributed to the result analysis and interpretation, commented on and approved the final paper.

**Competing Interests**

The authors declare that they have no known competing financial interests or personal relationships that could have appeared to influence the work reported in this paper.

**Acknowledgments**

We thank Patrick O'Keeffe, Dennis Finn, Jason Rich, and Matthew S. Roetcisoender for their assistance in the lab and field. This work was supported by the National Key Research and Development Program of China (2023YFC3008002), the Fundamental Research Funds for the Central Universities, Sun Yat-sen University (23ptpy91), and the National Science Foundation (NSF-AGS-1419614, NSF-AGS-1853050, NSF-AGS-1853354).

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
