# Peer review of "Uncertainties in temperature statistics and fluxes determined by"

_Atmospheric Measurement Techniques, 2024_

## Author Comment (AC1)

Anonymous Referee #1:

The eddy covariance technique has been widely used to measure turbulent fluxes between land and atmosphere. However, the influence of vibrations of the tower and mounting arms on temperature variances and fluxes still needs to be examined. Here, this manuscript examines 30-min average temperature variances and fluxes determined by eddy covariance systems including Campbell Scientific Anemometer Thermometry (CSAT3B) with closely co-located fine-wire thermocouples along with LI-COR $CO_2/H_2O$ gas analyzers at multiple heights above a sagebrush ecosystem. It is found that temperature variances and fluxes are underestimated by using sonic temperature (Ts) in comparison with fine-wire thermocouple temperature (Tc). The less energy of Ts spectra in the low-frequency range causes smaller variances and fluxes of Ts than Tc. The manuscript further investigated the potential causes for the discrepancies between variances and fluxes of Ts and Tc, and concluded that underestimated temperature variances and fluxes by using Ts are likely caused by wind-induced tower vibrations. These results are of great significance for improving our understanding when we calculate turbulent fluxes.

*Reply: Thank you very much for the comments! We have thoroughly revised the manuscript based on the comments.*

The following are a few minor comments I have on this manuscript:

1.    This study aimed to address a potentially important issue associated with eddy covariance measurements. It would be highly beneficial if the manuscript could include recommendations and to help explore and improve the potential issues caused by vibrations in future experiments.

*Reply: Thank you for the suggestions! We have included recommendations on the preferred tempature measurements to be used in investigating long-term $CO_2$ budget and energy balance closure. Our findings highlight the critical importance of accurate measurements of air temperature fluctuations in EC flux measurements. The inclusion of additional high-frequency temperature measurements using fine-wire thermocouples is strongly recommended for EC systems (lines 401-403).*

2.    This study used the measurements of Ts and Tc at three heights of 40.2, 23.0, and 12.8 m. Would the results remain consistent if data from other heights were utilized?

*Reply: Thank you for the comments! During the experiment, we installed the fine-wire thermocouples at six heights colocated with EC systems, including four levels on the tall tower (12.8, 15.8, 23.0 and 40.2 m) and two levels on the short tower (2.0 and 8.2 m). Figures R1, R2, R3, and R4 show the results for all six heights.*

*It was observed that the power spectra of u, w, and Ts deviated from the -5/3 power law in the high frequency range at 2.0 m and even at 8.2 m, most-likely caused by the influences from the roughness sublayer. The roughness length was determined to be a few centimeters at the experiment site (Finn et al., 2016). Also, there were structural differences between the tall tower and short tower setups. Therefore, due to these differences in tower structures and mounting*

*arms, as well as the potential influences from the roughness sublayer, measurements on the short tower were excluded from this study.*

*Furthermore, since there were small wind speed differences between 12.8 m and 15.8 m, their results were comparable; thus the measurement at 15.8 m were not included in this study.*

*We have clarified this in the revised manuscript (lines 128-129, and 146-149).*

[Figure]

*Figure R1. Comparison of temperature variances of Ts and Tc at the six heights of 40.2, 23.0, 15.8, 12.8, 8.2 and 2.0 m, respectively.*

[Figure]

*Figure R2. Comparison of sensible heat fluxes computed using Ts and Tc at the six heights of 40.2, 23.0, 15.8, 12.8, 8.2 and 2.0 m, respectively.*

[Figure]

*Figure R3. Mean normalized power spectra of u, w, Ts, and Tc at the six heights.*

[Figure]

*Figure R4. Mean normalized cospectra of the w-Ts, and w-Tc, at the six heights.*

3. How were the tower and sonic anemometers installed, was the tower guy wired, were the poles used to attach the sonic anemometers to the tower installed horizontally or vertically.

*Reply: Thank you for the comments! The 62-m tower was guyed at eight levels and the 10-m tower was guyed at one level. 3.6 m (12-ft) retractable square booms were horizontally braced to the 62-m tower to attach the sensors. The CSAT3s and IRGAs were mounted on 1-ft pipes, which were securely attached to the end of each boom (line 109-114).*

4. The results in Figure 7 are only shown for the measurements at 40.2 m. How about the results at the other two heights?

*Reply: Thank you for the comments! At 40.2 m, wind speeds varied from 0 to 18 ms$^{-1}$, and the influence of tower vibrations on temperature variances and fluxes becomes more pronounced during strong winds (refer to Figure 7 in the manuscript). Similarly, at lower levels like 23.0 m (see Figure R5), although wind speeds only ranged from 0 to around 16 ms$^{-1}$, the results were consistent with those observed at 40.2m. To maintain simplicity, we have not included the results for heights of 23.0 m and 12.8 m in the main text.*

[Figure]

*Figure R5. Ratios of half-hourly variances of $T_s$ and $T_c$ (left column) and covariances of w-$T_s$ and w-$T_c$ (right column) corresponding to the low, middle, and high frequency ranges, respectively, as a function of the mean wind speed ($\bar{u}$) at 23.0 m.*

5.    Line 258: "According to the comparison of the T_s and T_c spectra, the whole frequency domain can be divided into three regimes…" Probably replace the word "regimes" with "ranges" or "zones".

*Reply: Thank you for the suggestion! We have replaced the word "regimes" with "ranges" in the revised manuscript.*

---

## Author Comment (AC2)

Anonymous Referee #2:

Gao et al., utilizing multi-level co-located sonic anemometers and fine-wire thermocouples, compared the difference between temperature variances and fluxes derived from sonic anemometers and thermocouples and investigated the potential causes for the observed discrepancies. They found that temperature variances and fluxes determined from the sonic anemometers were underestimated in comparison with the counterparts determined from thermocouples, mainly attributed to the lower spectral energy in the low-frequency range. They concluded that the observed underestimation in temperature variances and fluxes determined from sonic temperature was likely caused by wind-induced vibrations of the tower and mounting arms.

*Reply: Thank you very much for the comments! We have thoroughly revised the manuscript based on the comments.*

The topic is interesting and is of great significance in the eddy covariance community. My comments are as follows:

1. My major comment pertains to the conclusions of this paper. The author concluded that the temperature variances and fluxes determined from sonic anemometer were underestimated which is likely attributed to the vibration and mounting arms especially in strong wind conditions. Besides, CO2 variances and fluxes were sensitive to such uncertainties. However, the paper does not specify which temperature product, the sonic-derived or thermocouple-measured, is more reliable. It would be beneficial if the author could provide recommendations on the preferred temperature product to be used in investigating long-term CO2 budget and energy balance closure. Can the author provide suggestions to reduce such uncertainties in scalar flux calculations?

*Reply: Thank you for the suggestions! We have revised the conclusion to provide recommendations on the preferred tempature measurements to be used in investigating long-term CO2 budget and energy balance closure. Our findings highlight the critical importance of accurate measurements of air temperature fluctuations in EC flux measurements. The inclusion of additional high-frequency temperature measurements using fine-wire thermocouples is strongly recommended for EC systems (lines 401-403).*

2. As shown in Figure 1, there are 8 levels of measurement on the tall tower and four levels of measurement on the short tower. Is there any reason for utilizing only three levels of measurement (12.8, 23.0, and 40.2 m)? Would the results remain consistent if measurements from other heights were used? It would be interesting to compare the data from 6 m (short tower) and 8.2 m (tall tower) since small wind speed differences are expected between these two heights. As a consequence, the influence of wind-induced vibrations of the tower and mounting arms on sonic temperature might be highlighted due to different tower structures and mounting arms.

*Reply: Thank you for the comments! During the experiment, we installed the fine-wire thermocouples at six heights colocated with EC systems, including four levels on the tall tower (12.8, 15.8, 23.0 and 40.2 m) and two levels on the short tower (2.0 and 8.2 m). Figures R1, R2, R3, and R4 show the results for all six heights.*

*It was observed that the power spectra of u, w, and Ts deviated from the -5/3 power law in the high frequency range at 2.0 m and even at 8.2 m, most-likely caused by the influences from the*

*roughness sublayer. The roughness length was determined to be a few centimeters at the experiment site (Finn et al., 2016). Also, there were structural differences between the tall tower and short tower setups. Therefore, due to these differences in tower structures and mounting arms, as well as the potential influences from the roughness sublayer, measurements on the short tower were excluded from this study.*

*Furthermore, since there were small wind speed differences between 12.8 m and 15.8 m, their results were comparable; thus the measurement at 15.8 m were not included in this study.*

*We have clarified this in the revised manuscript (lines 128-129, and 146-149).*

[Figure]

*Figure R1. Comparison of temperature variances of Ts and Tc at the six heights of 40.2, 23.0, 15.8, 12.8, 8.2 and 2.0 m, respectively.*

[Figure]

*Figure R2. Comparison of sensible heat fluxes computed using Ts and Tc at the six heights of 40.2, 23.0, 15.8, 12.8, 8.2 and 2.0 m, respectively.*

[Figure]

*Figure R3. Mean normalized power spectra of u, w, Ts, and Tc at the six heights.*

[Figure]

*Figure R4. Mean normalized cospectra of the w-Ts, and w-Tc, at the six heights.*

3. Figure 7 only shows the results at 40.2 m, can the author provide results at the other two heights?

*Reply: Thank you for the comments! At 40.2 m, wind speeds varied from 0 to 18 ms⁻¹, and the influence of tower vibrations on temperature variances and fluxes becomes more pronounced during strong winds (refer to Figure 7 in the manuscript). Similarly, at lower levels like 23.0 m (see Figure R5), although wind speeds only ranged from 0 to around 16 ms⁻¹, the results were consistent with those observed at 40.2m. To maintain simplicity, we have not included the results for heights of 23.0 m and 12.8 m in the main text.*

[Figure]

*Figure R5. Ratios of half-hourly variances of $T_s$ and $T_c$ (left column) and covariances of w-$T_s$ and w-$T_c$ (right column) corresponding to the low, middle, and high frequency ranges, respectively, as a function of the mean wind speed ($\bar{u}$) at 23.0 m.*